# Implications of nasal delivery of bromelain on its pharmacokinetics, tissue distribution and pharmacodynamic profile—A preclinical study

**Manu Sharma** ⓘ*, **Namita Gupta, Easha Pandey**

Department of Pharmacy, Banasthali Vidyapith, Jaipur, Rajasthan, India

* aasmanu2018@gmail.com, smanu@banasthali.in

**Data Availability Statement:** All relevant data are within the paper.

**Funding:** The authors received no specific funding for this work.

## Abstract

Asthma is a polygenic chronic inflammatory respiratory disease devastating the quality of life and state economies. Therefore, utilization of natural products as a therapeutic approach has attained wider consideration for development of novel drugs for asthma management. Bromelain, a mixture of natural bioactive cysteine proteases abundantly found in pineapple stem, has allured attention for its pharmacological activities. However, poor stability in gastric milieu, high dose and immunogenicity associated with prolonged use hinders its oral use. Therefore, need exists to explore alternative route of bromelain administration to achieve its plausible benefits. The present study investigated the preclinical prospects of nasal administration of bromelain on systemic bioavailability, tissue distribution and it's in vivo anti-histaminic, bronchodilator and anti-asthmatic activity in animal models. Pharmacokinetic studies revealed 1.43-fold higher relative bioavailability with faster absorption of bromelain on nasal administration at one-fourth oral dose. The enhanced cellular uptake and localization of bromelain in tissues of lung was observed significantly. Furthermore, faster onset and enhanced antihistaminic, bronchodilator and anti-asthmatic activity on bromelain's nasal administration signified faster absorption and higher in vivo stability of bromelain. Nasal administration significantly achieved decrease in level of oxidative and immunological markers along with restoration of antioxidant enzymes at considerably one-fourth dose administered orally. These findings distinctly manifested that nasal administration could be a substantial and effective route for bromelain delivery with enduring competency in asthma management.

## Introduction

Asthma is a most common chronic respiratory disease responsible for inflammation in lungs airways. It has mushroomed globally and impaired the quality of life of patients and their families [1]. According to WHO, India alone reports more than 20 million cases of asthma in a year. Asthma has been characterized by an exaggerated airway narrowing, airway obstruction, dyspnea and bronchospasm causing wheezing and difficulty in breathing witnessed in mild to severe disease [2]. Majority of pathophysiological alterations related to asthma are promoted by the activation of inflammatory cells and their accumulation, over expression of T cells,

**Competing interests:** The authors have declared that no competing interests exist.

abnormality of goblet cells and airway muscles. Excessive mucous production causes reversible or permanent changes in airway walls [3]. Allergen exposure to sensitized individuals trigger cascade of molecular events such as MAP (mitogen-activated protein) and NF-kB pathway. Thus, activation of biochemical and cellular events generates oxidative surrounding leading to impairment in lungs defense mechanism and accelerates the devastation of macromolecules [4, 5].

The available treatments either handle symptoms (bronchodilators) or prevent the symptoms (steroids, anti-histaminic and anti-inflammatory agents). Among the treatments, inhaled glucocorticoids are first line therapy recommended for asthma management [6]. However, prolonged corticosteroid therapy is associated with several putative systemic side effects which reduce the treatment compliance [7]. Moreover, patients with severe respiratory diseases like asthma, chronic obstructive pulmonary disease has shown the resistance to glucocorticoids therapy due to abnormalities in glucocorticoid receptor signaling pathways [8]. Patients are also often reluctant for long term therapies with a fear of serious side effects and may opt a variety of alternative approaches to treat their asthma. Furthermore, individuals suffering with asthma have potential to devastate health systems and state economies due to higher cost of therapies [9]. Thus, need exists to find more economical, safer and effective therapy with minimal systemic side effect for asthma treatment. This has attracted the attention of pharmaceutical scientists, clinicians as well as patients towards herbal therapies to reduce or eliminate the usage of synthetic drugs and inhalers for asthma management [10].

Currently, cysteine protease families are receiving extensive attention because of their cost effective, presumed safety and potential therapeutic effects in clinics [11, 12]. Bromelain, a complex mixture of proteolytic cysteine proteases obtained from stem of pineapple (*Ananas comosus*) is a food supplement exhibiting a variety of pharmacological activities. Various studies substantiate the bromelain's mucolytic [13], wound healing [14], fibrinolytic [15], antiedematous [16], antithrombotic [17], anti-inflammatory [18], antioxidant [19], anticancer [20] as well as immunomodulatory activity [21]. Bromelain significantly decreases COX-2 and PGE-2 expression and lower the cytokines (IL-1β, IL-6 and TNF-α) secretion from immune cells in inflammatory conditions [22, 23]. Several reports evidently demonstrated the efficacy of bromelain in reducing airway reactivity and susceptibility to irritants by decreasing markers (CD19[+] B cells, CD4[+] and CD8[+] T lymphocytes) of lung inflammation in ovalbumin induced model of allergic airway disease [24, 25]. However, bromelain's instability in gastric milieu limits its pharmacological applications on oral administration [18]. This necessitates the exploration of alternative route of bromelain administration to achieve its plausible health benefits.

Nasal route has been found effective and promising for gastro-sensitive drugs to achieve better systemic bioavailability compared to oral administration. It is a non-invasive highly vascularized route of drug administration bypassing first pass metabolism and avoiding presystemic elimination of GIT tract [26]. Thus, nasal administration achieves higher localized and systemic therapeutic effect with smaller dose size and lesser dosing frequency compared to oral route [27]. Therefore, present study was undertaken to evaluate the systemic bioavailability, tissue distribution and therapeutic efficacy of bromelain in acute bronchospasm and OVA induced asthma in guinea pig model via nasal administration in comparison to oral.

## Materials and methods

### Chemicals and reagents

Bromelain, casein, tyrosine, ovalbumin (OVA) and trichloro acetate (98.0%) were procured from Himedia laboratory Pvt. Ltd., Mumbai, India. All other chemicals and solvents were of analytical grade and were used without further purification. Double distilled water was used

throughout the study. TNF-α, IL-5 and IgG ELISA kits were used as per the manufacturer's guidelines (Bioassay technology laboratory, China) respectively.

## Experimental animals

Young adult Wistar rats, weighing 180–220 g and male guinea pigs (250–300 g) used in the study were obtained from animal house of Department of Pharmacy, Banasthali Vidyapith. Animals were nurtured in well ventilated cages at standard conditions of temperature (23 ± 2˚C), relative humidity (60–70%) and a controlled 12 h light/dark cycle. Animal bedding was changed daily to maintain hygienic conditions. Standard commercial chow diet and water *ad libitum* throughout the study was provided to test animals. The study was approved by Banasthali Vidyapith Institutional Animal Ethical Committee (574/GO/ReBi/S/02/CPCSEA) in accordance of CPCSEA guidelines of Ministry of Animal Welfare, Government of India. All experiments were performed in accordance with the guidelines established by Institutional Animal Ethical Committee (IAEC) for the care and use of laboratory animals. At the end of the experiments, euthanasia was performed using thiopental Sodium (100mg/Kg) + Xylazine (39 mg/Kg) I.V. and carcass was incinerated at incinerator facility.

## Pharmacokinetic studies

Twenty four Wistar rats were randomly distributed into two groups with twelve animals in each group. Animals of group I received bromelain solution orally (400 μl, 40 mg/Kg body weight) by oral gavage while second group received 30 μl, bromelain solution (10 mg/ Kg body weight) in each nostril in 30 s with the help of micropipette with 0.1 mm internal diameter at delivery place. Nasal dose was administered using intransal grip, each rat was restrained twice and held with their neck parallel to the floor while administering nasal dose. Blood samples (300 μl) were collected from rat tail vein in heparinised coated tubes at pre-dose (0.0), 0.5, 1, 2, 4 and 6 h post dose of bromelain solution. Plasma was collected by centrifuging samples at 5000 rpm at 4˚C for 10 min and stored at -80˚C until further analyzed for bromelain's proteolytic activity present respectively [17].

## Tissue distribution study

Sixty healthy wistar rats were randomized into two groups (n = 30). Rats of group I received bromelain solution orally (40 mg/kg body weight) while group II received bromelain solution nasally (10 mg/kg body weight) in each nostril with the help of micropipette with 0.1 mm internal diameter respectively. Six animals per group were sacrificed at 0.5, 1, 2, 4 and 6 h post dosing. Immediately after sacrificing animals, lung and liver was collected, washed, blotted dry on a tissue paper, and weighed. Tissue samples were minced with phosphate buffer saline (PBS) and homogenized to fine paste in a tissue homogenizer (Remi, Mumbai, India). Homogenates were suitably processed by addition of protein precipitating agents and centrifuged for 20 min at 12,000 rpm. Clear supernatant obtained was analyzed for bromelain's proteolytic activity [17].

## Pharmacodynamics studies

**Histamine induced bronchospasm.** Histamine induced bronchospasm model was employed to evaluate *in vivo* anti-histaminic and bronchodilator activity of bromelain in guinea pigs [28]. Forty-five overnight fasted animals were randomly divided into three groups (n = 15): Group I: sham group received PBS; group II and III treated with bromelain solution orally (40 mg/kg body weight) and nasally (10 mg/kg body weight) respectively. Animals were

challenged with histamine dihydrochloride aerosol (0.5% w/v) using a nebulizer at a pressure of 300 mm Hg in an air tight Plexiglass chamber ($24 \times 14 \times 24$ cm) for 1 min. Consequently after histamine exposure, animals were closely observed to record the pre-convulsive dyspnea time and convulsion time for 9 minutes. The pre-convulsive dyspnea (PDT), i.e., the time of aerosol exposure to the onset of difficulty in breathing leading to the appearance of convulsions, and time for recovery (RT) i.e., the time taken by animals with pre-convulsive dyspnea of respective treated group to recover when placed in fresh air were recorded and compared with saline treated group [29]. The bronchospasm activity was recorded at different time intervals (0.5, 1, 2, 4 and 6 h; 3 animals at each time point) in pre-treated animals of respective group. Then, animals receiving histamine exposure at 6 h of post dose were sacrificed to collect blood and tissue samples and evaluated for oxidative stress and immunological biomarkers [17].

**OVA induced asthma.** *Sensitization and antigen challenge*. Fifteen healthy male guinea pigs (250–300 g) were sensitized by intraperitoneal injections of OVA and alum solution (150 µg ovalbumin and 100 mg aluminum hydroxide emulsified in 1 ml of normal saline) on day 1 and 7 respectively. Afterward on 14[th] day, a booster dose of OVA solution was injected to complete the sensitization phase. Naïve animals (n = 5) were sham sensitized with 100 µl normal saline similar to OVA sensitized animals.

*Assessment of in vivo efficacy*. OVA sensitized animals were arbitrarily divided into three groups (n = 5) on completion of sensitization phase. Animals of group I received normal saline and group II and III received bromelain solution orally (40 mg/ kg) and nasally (10 mg/ kg) respectively for 5 days. Naïve animals (group IV) received PBS. After receiving 5 days of treatment, animals of respective group were exposed with 1% histamine dihydrochloride solution in a clear plexiglass histamine chamber. The pre-convulsive dyspnea (PDT) and time for recovery (RT) were recorded and compared with saline treated group [29]. Animals were closely observed for survival and physiological responses throughout 20 days of study.

Subsequently, animals were sacrificed after completion of study to collect blood samples by cardiac puncture and organs like lung, liver and trachea. Whole blood collected was assessed for hematological parameters like total leukocytes, hemoglobin, lymphocytes, eosinophils and neutrophils. Serum separated from whole blood was collected and stored at -80 ℃ until used to measure immunological parameters like TNF-α, IL-5 and IgG level using ELISA kits as per the manufacturer's guidelines (Bioassay technology laboratory, China). Successively, tissue homogenates were prepared in potassium chloride solutions (10% w/v) to evaluate for antigen specific response by quantitating level of oxidative stress markers like lipid peroxidation, carbonyl content, myeloperoxidase activity, reduced glutathione, superoxide dismutase, catalase, nitric oxide and EPO level [18, 30].

*Bronchoalveolar fluid analysis*. Trachea of sacrificed animals immediately after blood collection was carefully exposed and cannulated. Bronchoalveolar lavage was executed five times by infusing normal saline (2 ml) into lungs via cannula and aspirated after gentle massage. Samples collected were pooled to determine total number of cells/ ml using Neubauer hemocytometer. Slides were stained for 15 min with Giemsa stain to count for minimum of 200 cells/ slide. Separated BAL fluid was stored at -80 ℃ until used to measure immunological parameters like TNF-α, IL-5 and IgG level using ELISA kits as per the manufacturer's guidelines (Bioassay technology laboratory, China) as well as evaluated for oxidant and antioxidant makers' level.

**Histopathological studies.** Isolated lung was diced into 0.5 cm cubes and fastened in neutral buffered formalin (10%), processed and wedged in paraffin for histological evaluation. Thin slices of lung were cut, secured on glass slides, deparaffinized and stained with haematoxylin-eosin before capturing images under microscope for observing histopathological changes.

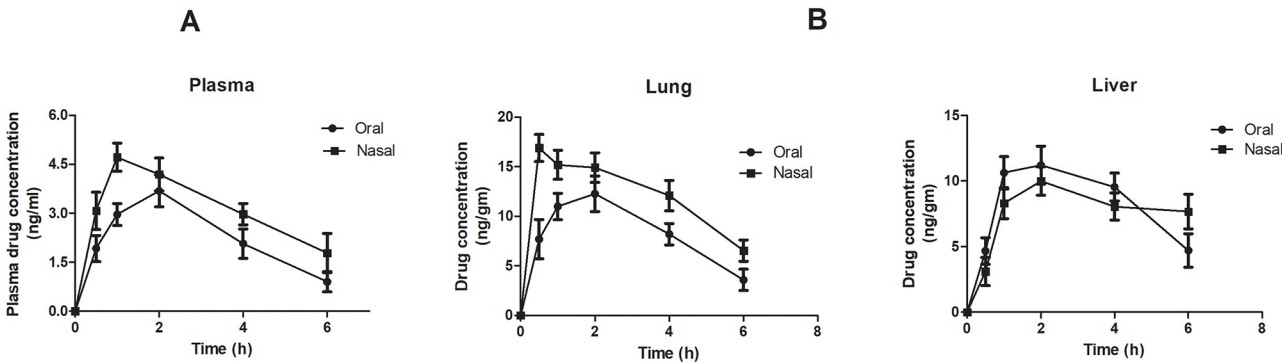

**Fig 1.** (A) Plasma concentration-time profile of bromelain and (B) Tissue distribution profile of bromelain in lungs and liver after oral (40 mg/Kg) and nasal (10 mg/Kg) administration of bromelain solution respectively.

## Statistical analysis

Data are presented as mean ± standard deviation. One-way analysis of variance followed by Bonferroni test was performed for multiple comparisons of related data using GraphPad Prism software version 5.00, USA to determine the statistically significant difference at $p < 0.05$.

## Results and discussion

### Pharmacokinetic study

Plasma concentration-time profile of bromelain after single post oral (40 mg/Kg) and post nasal (10 mg/kg) administration in wistar rats have been presented in Fig 1A respectively. The obtained data was evaluated by non-compartment pharmacokinetics analysis utilizing Win-Nonin (version 5.1, Pharsight Corporation, Mountain view, USA). Pharmacokinetic parameters computed are summarized in Table 1. The Cmax was found 1.28-fold higher ($p < 0.05$) on post nasal administration whereas 2-fold decrease in Tmax was observed compared to oral administration. Significant ($p < 0.05$) increase in $t_{1/2}$ (1.82-fold) and MRT (1.64-fold) respectively was observed on nasal administration of bromelain. Meanwhile $AUC_{0-24h}$ was also increased 1.43-fold for bromelain administered nasally compared to oral drug. The results confirmed that nasal administration facilitated the rapid systemic absorption of bromelain from nasal cavity and avoided pre-systemic elimination of drug from gastrointestinal tract by avoiding denaturation in gastric milieu and proteolytic degradation. The enhanced systemic

**Table 1. Pharmacokinetic parameters of bromelain after oral (40 mg/kg) and nasal (10 mg/kg) administration of bromelain solution.**

| Parameter | Oral drug | Nasal Drug |
|---|---|---|
| Cmax (ng ml$^{-1}$)* | 3.68 ± 0.34 | 4.71 ± 0.41 |
| Tmax (h)* | 2.00 ± 0.11 | 1.00 ± 0.12 |
| Ke (h$^{-1}$)* | 0.35 ± 0.11 | 0.19 ± 0.08 |
| $t_{1/2}$ (h)* | 1.97 ± 0.32 | 3.58 ± 0.48 |
| MRT (h)* | 3.00 ± 0.49 | 4.93 ± 0.77 |
| AUC (ng h$^2$ ml$^{-1}$)* | 13.37 ± 1.22 | 19.07 ± 2.21 |

Values are expressed as mean ± SD, n = 3;

* indicates $p < 0.05$ level of significant difference

absorption with nasal administration might be contributed by M cells in nasal associated lymphoid tissues (NALT) [26]. These in vivo studies confirmed that bioavailability of bromelain can be increased via nasal administration.

## Tissue distribution

Fig 1B represents distribution pattern of bromelain in lungs and liver at different time interval after single post oral (40 mg/kg) and nasal dose (10 mg/kg) of bromelain solution. Peak drug concentration in lung and liver was achieved between 0.5 to 2 h. Bromelain administered intra-nasally achieved remarkably ($p < 0.05$) higher concentration in lungs compared to oral within half an hour indicating higher localization of bromelain in lungs. Simultaneously, higher drug concentration in lungs with progression of time also confirmed protection offered by nasal route to drug metabolism (Fig 1B). Lower drug concentration in liver with progression of time too assured that nasal administration bypasses bromelain degradation or denaturation encountered in acidic milieu of stomach as well as the first pass metabolism of drug. The results further confirmed the efficacy of nasal route in providing protection to bromelain and increase the availability of bromelain at the site of action i.e., lungs.

## Acute bronchospasm

Histamine was used as spasmogenic agent to induce pre-convulsive dyspnea i.e., to provoke bronchospasm, rigorous bronchial airway constriction ensuing hypoxia which evokes convulsion in animals [31]. Frequent bronchospasm onset, jerks, severe spasm as well as death was observed in control group animals. Treated groups showed no mortality throughout the study. Group receiving bromelain via nasal route showed significantly increased bronchospasm onset time by 26.94% (from 1.00 ± 0.03 min to 1.23 ± 0.02 min) and 21.78% (from 1.05 ± 0.02 min to 1.23 ±0.02 min; $p < 0.05$) compared to saline and oral bromelain treated group respectively. The percentage recovery time after histamine induced bronchospasm for nasal bromelain treatment decreased 58.20% (from 1.34 ± 0.05 min to 0.56 ± 0.03 min; $p < 0.05$) compared to oral treatment (Fig 2). The improvement in bromelain activity via nasal route might be attributed to bypass of the first pass metabolism, avoidance of the presystemic elimination via GIT tract and rapid drug absorption from highly vascularized nasal surface [32]. Lower pre-convulsive dyspnea efficacy in orally treated group might be attributed to degradation of drug in acidic environment leading to reduction in its therapeutic efficacy. The remarkably higher bronchospasm onset time and percentage protection along with prevention of convulsion and faster recovery time was observed initially from half an hour to 2 h after nasal administration compared to oral. The faster absorption of drug through nasal delivery might be contributing to higher protection against spasmogenic potential of histamine.

**Oxidative stress and immunological biomarkers.** Histamine challenge mediated oxidative stress led to bronchoconstriction by stimulating H1 receptors on airway smooth muscles. In clinical practices, free radical mediated lipid peroxidation and protein carbonylation are main markers of oxidative load in patients with bronchospasm. Bromelain was found efficient in reducing the extent of lipid peroxides and protein carbonyl groups in lung, liver and trachea significantly in case of both orally and nasally treated group compared to saline treatment ($p < 0.05$). Nasally treated group showed considerable reduction of lipid peroxide and protein carbonyl level in lung, liver as well as trachea compared to orally treated (Fig 3). This might be attributed to protection of proteolytic activity of bromelain offered by nasal route as compared to oral route [25, 29].

Animals challenged with histamine showed remarkable elevation in lymphocytes (1.68-fold), neutrophils (1.86-fold) and leucocytes (3.17-fold), essential regulators of the

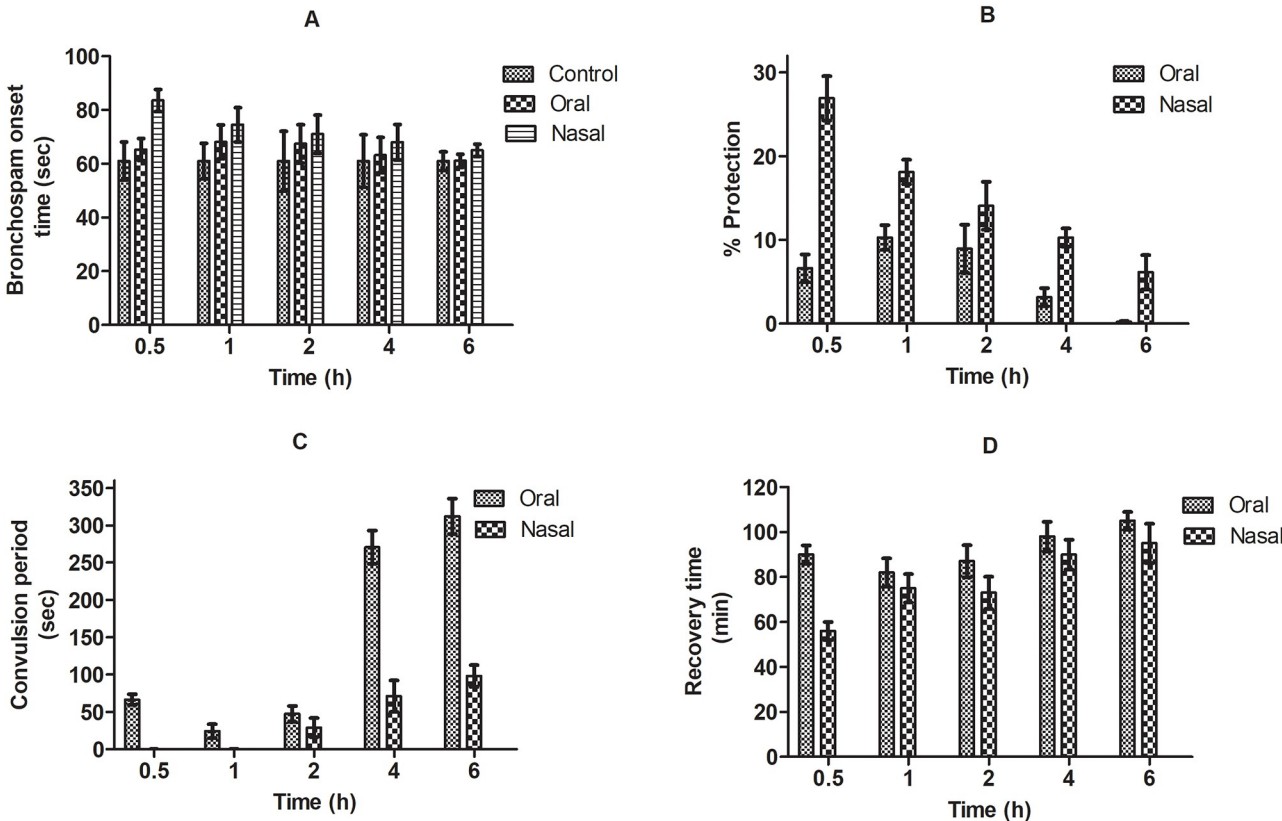

**Fig 2.** Effect of oral and nasal administration of bromelain on (A) bronchospasm onset time, (B) percentage protection, (C) convulsion period and (D) recovery time respectively during histamine induced acute bronchospasm.

immune response compared to naïve control group. Elevated level of lymphocytes and neutrophils initiated a cascade of events in lungs facilitating secretion of Th2-type cytokines, including IL-5, IL-13 and IL-4. The augmented level of cytokines further endorses mucus production, inflammatory actions and edematous conditions along with stimulation of profibrotic transformation of TGF-β, which is expressed in alveolar macrophages, fibroblasts and endothelial cells of lung [33]. However, a significant decrease in total leucocytes count, lymphocytes and neutrophils level was observed in bromelain treated groups emphasized reduced inflammation of lungs compared to saline. Nasal delivery of bromelain assured higher reduction of lung inflammation by significantly reducing the level of immunological biomarkers i.e., leucocytes (1.42-fold), lymphocytes (1.90-fold) and neutrophils (1.23-fold) compared to oral drug (Fig 3).

## Effect of bromelain on OVA induced airway inflammation

Airway hyperresponsiveness is one of the distinct features of asthma harmonized by diverse set of genes implicated in altering immunity and airway inflammation [34]. Histamine provocation has been broadly used aid in diagnosis of allergic respiratory disorders to evaluate airway responsiveness [35]. Ovalbumin sensitization induced significant airway hyperresponsiveness to histamine challenge in guinea pigs.

Various clinical parameters like bronchospasm onset, intensity of spasm and persistent episodes of convulsions were observed during study to evaluate the treatment efficacy on airways hyperresponsiveness against histamine inhalation. OVA sensitized animals showed

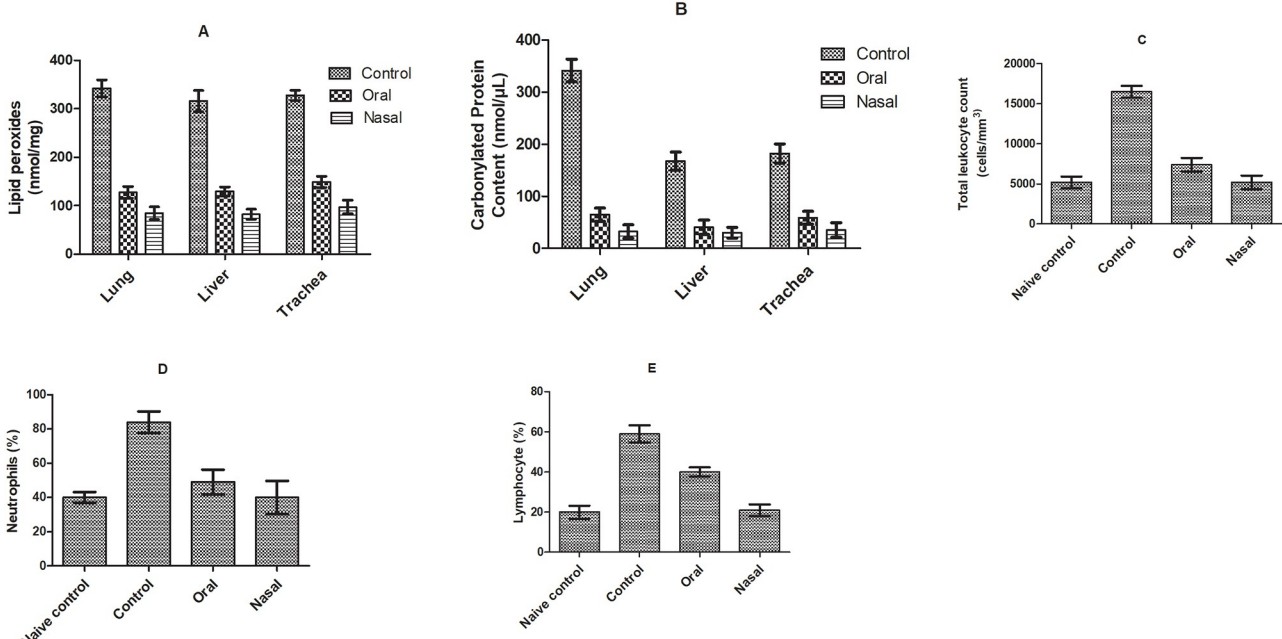

**Fig 3.** The effect of bromelain solution administered orally and nasally on oxidative stress markers (A) LPO level, (B) carbonylated protein content; and hematological immune response modulators (C) total leukocyte count (D) neutrophil (E) lymphocyte in lung, liver and trachea respectively in acute bronchospasm model.

considerably higher bronchial hyperresponsiveness in response to histamine compared to normal saline treated animals ($p < 0.05$). Notable elevation in duration for histamine induced pre-convulsive dyspnea by 9.83% and 29.50% (from $1.00 \pm 0.04$ min to between $1.08 \pm 0.07$ min, $1.22 \pm 0.12$ min; $p < 0.05$) whereas decline in recovery time after pre-convulsive dyspnea by 47.73% and 53.31% (from $2.87 \pm 0.54$ min to $1.5 \pm 0.07$ min, $2.87 \pm 0.54$ min to $1.34 \pm 0.07$ min; $p < 0.05$) was observed in animals treated by oral and nasal dose of bromelain respectively. The significant delay in bronchospasm induction and jerks were observed in nasally treated animals as compared to positive control (sensitized, saline treated) and oral drug treated animals ($p < 0.05$) (Fig 4). The protection of proteolytic activity of bromelain might be contributing higher systemic bioavailability and efficacy of drug via nasal route.

The increase in the relative weight of liver ($47.04 \pm 1.23\%$) and lungs ($53.60 \pm 2.02\%$) in OVA sensitized animals compared to control group was observed (Fig 5). The increased weight of lung and liver suggested that OVA exposure aggravated microvascular infiltration and oedema resulting swelling of respective organs. Significantly diminished relative weight of lung and liver was observed on oral and nasal treatment of bromelain ($p < 0.05$) compared to OVA sensitized group (Fig 5). The results indicated that bromelain lessened the tissue damage and oedema in lungs due to its antioxidant and anti-inflammatory activity [36].

OVA sensitization facilitated the remarkable elevation of immune response regulators i.e., total leukocyte count (3.86-fold), lymphocytes (1.74-fold), eosinophils (2.25-fold), neutrophil (2.04-fold) and hemoglobin (1.46-fold) compared to naïve control animals (Fig 6). Simultaneously, elevated level of eosinophil count established the allergen induced allergic inflammation associated with airway hyperresponsiveness [37]. Asthma related pulmonary hypoxia might have evoked the hemoglobin synthesis which increased serum hemoglobin level in control group [38]. Nasal delivery of bromelain significantly ($p < 0.05$) reduced total leukocyte count, lymphocyte, platelet count, neutrophils and eosinophils count to relieve the

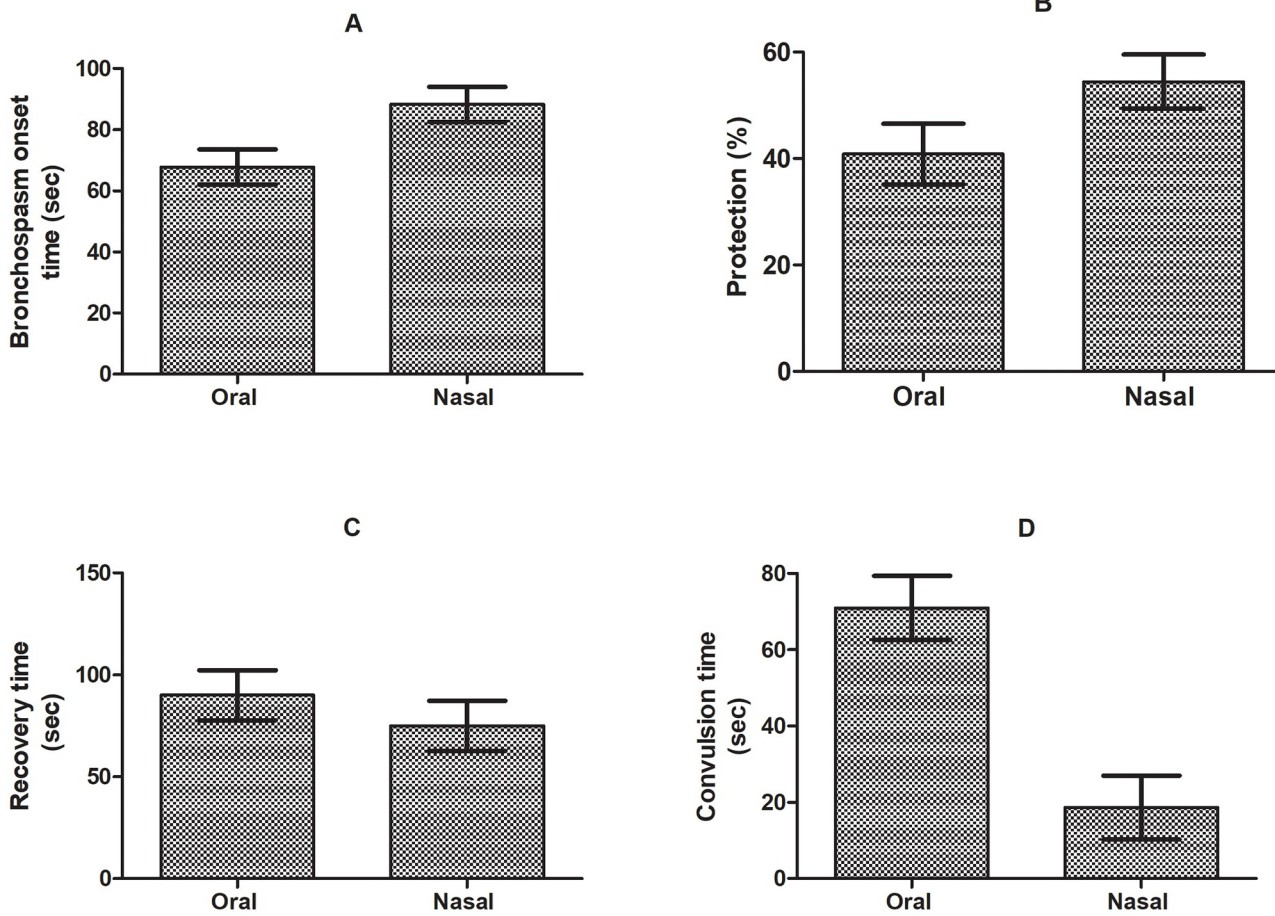

**Fig 4.** Effect of oral and nasal treatment on (A) bronchospasm onset time, (B) percentage protection, (C) recovery time and (D) convulsion time respectively in ovalbumin induced asthma model.

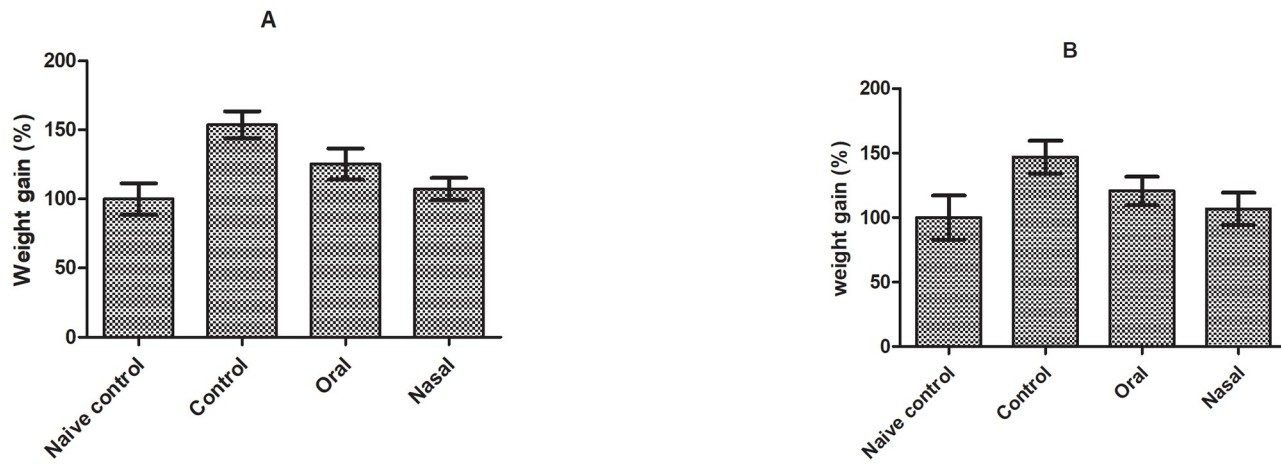

**Fig 5.** Effect of different treatments on weight of (A) lung and (B) liver in OVA induced asthma model.

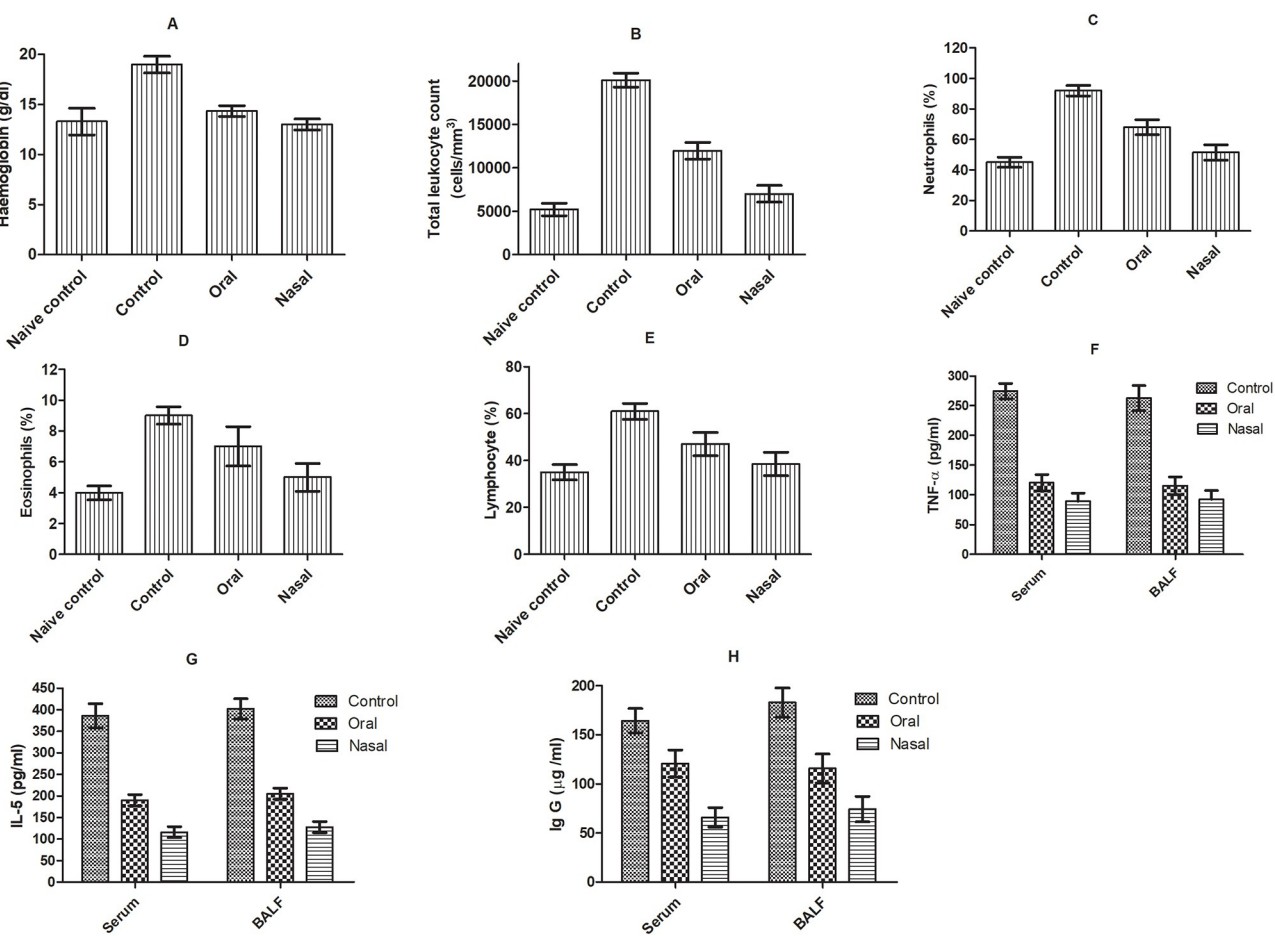

**Fig 6.** Effect of oral and nasal bromelain treatment on hematological and immunological biomarkers (A) hemoglobin, (B) total leukocyte count, (C) neutrophils, (D) eosinophils, (E) lymphocyte, (F) TNF–α, (G) IL-5, (H) Ig-G respectively in OVA induced asthma model.

inflammatory state in comparison of orally bromelain and saline treated animals respectively (Fig 6). The reduction of leucocyte subpopulation systemically suggested that bromelain activity is focused at all inflammatory sites.

Elevation in NO level stipulated the overexpression of inducible nitric oxide synthase (iNOS) and prostaglandins genesis in OVA sensitized control [39]. Correspondingly, escalated level of TNF-α, Ig-G and IL-5 in serum facilitates the promotion of neutrophils migration to tissues, NO synthesis and reactive oxygen species production [40]. However, treatment with oral and nasal bromelain showed significant reduction of WBC count, MPO, NO, Ig-G and cytokines level (IL-6 and TNF-α) respectively compared to saline (Figs 6 and 7). Thus, bromelain can inhibit progression of allergic disease by diminishing T cell driven inflammation. Bromelain's inhibitory effect on $NF_{KB}$ overexpression and iNOS along with improved expression of $Nrf_2$ pathway had imparted its beneficial antioxidant and anti-inflammatory effects for treatment of inflammatory conditions like asthma [41]. The enhanced anti-inflammatory activity of bromelain on nasal administration might be attributed to improved proteolytic activity by avoiding the pre-systemic elimination from GIT tract. Moreover, nasal route offered large surface area for absorption with high vascularity. It is assumed that bromelain's inhibitory effect on cytokine storm might be attributed by modulating Th2 cells and reducing pro-inflammatory cytokines, chemokines, NO production through $NF_{KB}$ and MAP pathways stimulated in bronchial epithelial cells [41].

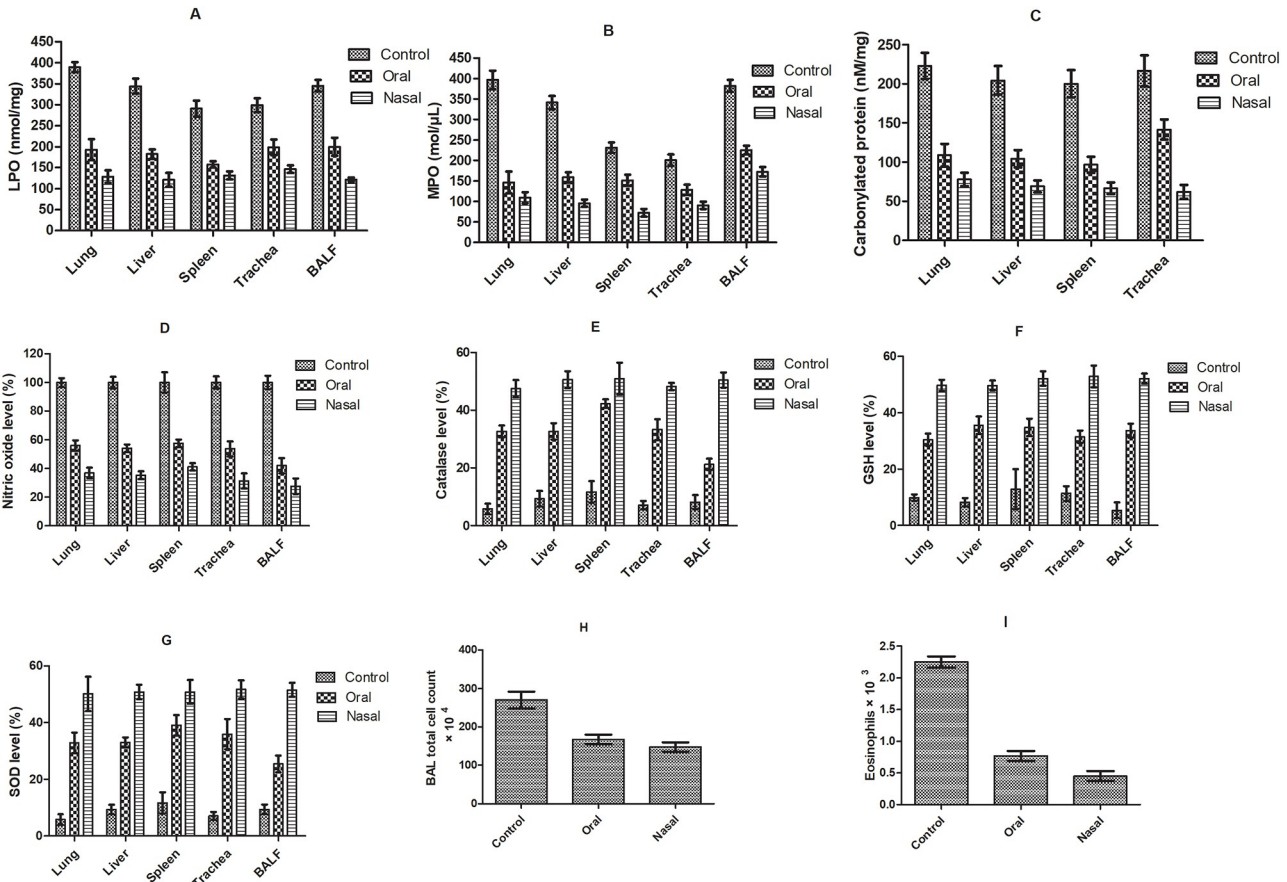

**Fig 7.** Effect of different treatments in OVA induced asthma model on (A) LPO, (B) MPO activity, (C) carbonylated protein content, (D) NO (%), (E) catalase (%), (F) GSH (%), (G) SOD (%) in lung, liver, spleen, trachea and BALF respectively; (H) BAL total cell count and (I) eosinophil count respectively.

Analysis of BAL fluid and supernatants of tissue homogenates showed consistent increase in level of oxidative markers (LPO, MPO, EPO, NO) with decrease in antioxidant enzymes (SOD, and CAT) and GSH in ovalbumin challenged group. Decreased levels of SOD, CAT and GSH with increased level of LPO and carbonylated protein in OVA challenged animals showed imbalance between free radical production and antioxidant defense ensuing oxidative cellular stress which facilitated airways inflammation and asthma severity [18] (Fig 7). However, oral as well as nasal bromelain treated groups manifested remarkable mitigation of oxidative stress by depleting LPO, MPO, NO, EPO, carbonylated protein level and total cell count in BAL fluid respectively compared to OVA sensitized control ($p < 0.05$). The results indicated that nasal bromelain was more effective in restoring SOD, catalase and GSH level compared to oral bromelain ($p < 0.05$) (Fig 7). The marked suppression of oxidative stress markers, amplification of antioxidant defense and anti-inflammatory activity confirmed the improved potential of bromelain in management of airway hyperresponsiveness via nasal route [42].

## Histopathological evaluation

Histopathological microphotographs of lung of naïve, normal saline, bromelain treated animals via oral and nasal route respectively are shown in Fig 8. The architecture of lung in naïve animals showed clear alveolar sacs with no gathering of cells in bronchiole's region. However,

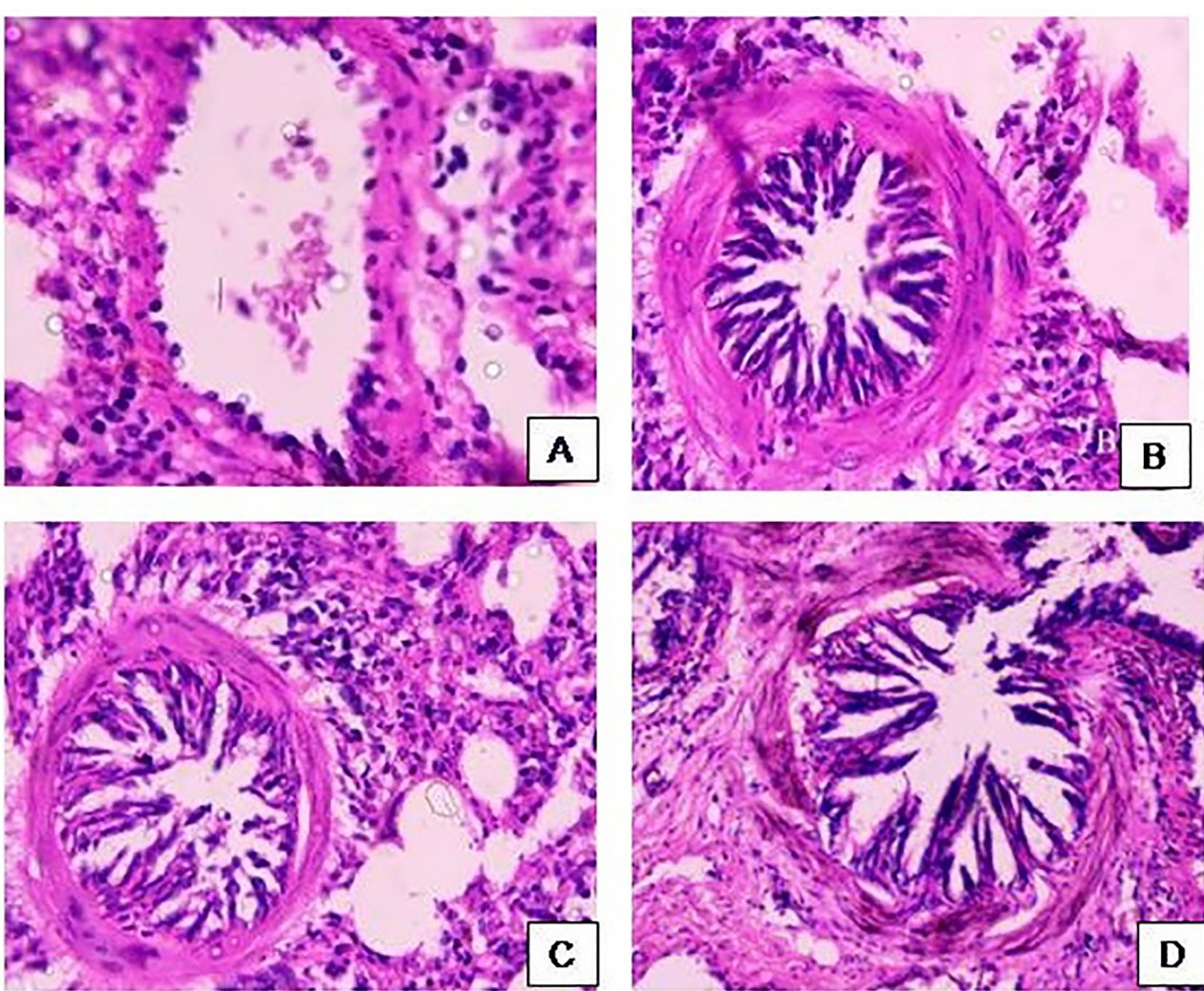

**Fig 8.** Histopathological images of lung of (A) naïve control, (B) OVA sensitized control, (C) orally treated, and (D) nasally treated animals.

OVA sensitized group showed substantial deposition of cells in bronchioles, blood vessels and alveolar regions indicative of extreme infiltration of cells (eosinophil, lymphocyte and mono-cyte) leading to constriction of alveolar sacs and tissue damage. OVA exposure mitigated the oxidative burden which might be contributing in thickening of airway smooth muscles and basement membrane. The epithelial cell hyperplasia observed was recognized as lung remodel-ing. However, orally and nasally bromelain treated groups showed less accumulation of cells around the bronchioles as well as less thickening in alveolar septa compared to control group. Lung microphotographs of animals receiving bromelain nasally showed loosely confined structure confirming lesser infiltration of cells in bronchioles region compared to oral treat-ment [43].

## Conclusions

The outcome of present study conferred that bromelain retained its therapeutic potential sig-nificantly higher via nasal route compared to oral. Nasal administration improved systemic bioavailability of bromelain as well as inhibited and modulated critical components of airway

hyperresponsiveness. The astounding in vivo antioxidant, anti-inflammatory and anti-asthmatic activity via nasal delivery of bromelain at lower dose compared to oral was noticed. All these positive attributes make nasal route amenable for bromelain use in management of asthma. However, more meticulous assessment is required in fundamental biological models to conceive clinical trials properly.

## Acknowledgments

The authors acknowledge the Department of Pharmacy, Banasthali Vidyapith (Rajasthan, India) for providing facilities to undertake this research.

## Author Contributions

**Conceptualization:** Manu Sharma.

**Data curation:** Manu Sharma, Namita Gupta, Easha Pandey.

**Formal analysis:** Manu Sharma.

**Investigation:** Manu Sharma, Namita Gupta, Easha Pandey.

**Methodology:** Manu Sharma.

**Writing – original draft:** Manu Sharma.

**Writing – review & editing:** Manu Sharma, Namita Gupta.

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
