## [Decision Letter · Decision Letter 0]

25 Jul 2022

PONE-D-22-16222Implications of nasal delivery of bromelain on its pharmacokinetics, tissue distribution and pharmacodynamic profilePLOS ONE

Dear Dr. Sharma,

Thank you for submitting your manuscript to PLOS ONE. After careful consideration, we feel that it has merit but does not fully meet PLOS ONE’s publication criteria as it currently stands. Therefore, we invite you to submit a revised version of the manuscript that addresses the points raised during the review process.

We look forward to receiving your revised manuscript.

Kind regards,

Lisa Ann Tell

Academic Editor

PLOS ONE

Journal Requirements:

Additional Editor Comments:

Dear Authors:

Please make modifications to this manuscript to indicate that it is a preliminary study. The number of animal subjects overall and animal subjects per time point are small which doesn't really allow for assessment of differences. Please address this with your revision.

Reviewers' comments:

Reviewer's Responses to Questions

**Comments to the Author**

1. Is the manuscript technically sound, and do the data support the conclusions?

Reviewer #1: Yes

Reviewer #2: Yes

2. Has the statistical analysis been performed appropriately and rigorously? 

Reviewer #1: Yes

Reviewer #2: Yes

3. Have the authors made all data underlying the findings in their manuscript fully available?

Reviewer #1: Yes

Reviewer #2: Yes

4. Is the manuscript presented in an intelligible fashion and written in standard English?

Reviewer #1: Yes

Reviewer #2: Yes

5. Review Comments to the Author

Reviewer #1: The PK and tissue distribution of bromelain were conducted and compared in rats following oral and nasal administration, and the pharmacodynamic profile was determined in guinea pigs. The present study is within the scope of PLOS ONE. HOWEVER, the writing elements need further improvement to enhance the quality of the manuscript, and copy editing is beyond the scope of a reviewer’s charge. The authors should pay more attention to further polishing this manuscript.

Some of my specific comments are listed below:

1. The animals used in this study (rats and guinea pigs) should be added to both the title and abstract. Otherwise, readers will be misled into thinking that this study was conducted on human patients.

2. 2.3. Pharmacokinetic studies: To this reviewer’s knowledge, six rats were not enough in a PK study. Additionally, where were the blood samples collected? How many milliliters of blood were collected each time? All these details should be given in the revision.

3. Three animals per group were sacrificed at 0.5, 1, 2, 4 and 6 h post dosing. Likewise, three rats per time point were not sufficient. These data may not be sufficient to describe inter-individual differences.

4. Animals were exposed to an aerosol of … The authors should pay more attention to English.

5. Table 1: Which parameters were statistically significantly different (P<0.05)?

6. Figure 1B: This reviewer suggests the authors change the style of Figure 1B to Figure 1A.

Reviewer #2: Abstract – The meaning of “unforeseeable prophylactic hypersensitivity due to its denaturation or agglomeration hinders its oral use” is unclear.

Section 2.4 – For the oral dose of bromelain solution (40 mg/kg), what is the volume of administration? Is the drug in a solution or in a suspension? What is the vehicle of the solution? Is there any pH adjustment?

The same questions can be applied to the nasal dose solution (10 mg/kg).

How is the nasal dose administered? Are the animals constrained in a chamber? Is the nasal dose administered over a duration?

Table 1, the increased in AUC looks like to be caused by a decrease in elimination? It may not be due to increase in bioavailability alone. The improved in efficacy may be due to higher drug concentration in the lung at 0.5 hour.

Figure 1 – it looks like the substantial difference in the lung concentration is only observed at 0.5 hour. In Figure 1, labelling with “A” and “B” and “a” and “b” is confusing.

6. PLOS authors have the option to publish the peer review history of their article (what does this mean?). If published, this will include your full peer review and any attached files.

Reviewer #1: **Yes: **Fan Yang

Reviewer #2: No

---

## [Author Response · Author response to Decision Letter 0]

8 Sep 2022

Academic Editor Comments

Comment 1: Please ensure that your manuscript meets PLOS ONE's style requirements, including those for file naming. The PLOS ONE style templates can be found at 

Response: We ensure that our revised manuscript meets PLOS ONE’s style requirements, including those for file naming.

Comment 2: To comply with PLOS ONE submissions requirements, in your Methods section, please provide additional information regarding the experiments involving animals and ensure you have included details on (1) methods of sacrifice, (2) methods of anesthesia and/or analgesia, and (3) efforts to alleviate suffering.

Response: Additional information regarding the experiments involving animals have been included in Method section in detail in revised manuscript as per suggestion made by Academic Editor. 

Comment 3: Please make modifications to this manuscript to indicate that it is a preliminary study. The number of animal subjects overall and animal subjects per time point are small which doesn't really allow for assessment of differences. Please address this with your revision.

Response: As per editor’s suggestion we have made changes in manuscript title and content to indicate that it is a preliminary study. We have performed further studies to derive data on sufficiently large number of animal subjects overall and animal subjects per time point for assessment of differences. The results have been included in revised manuscript.

Response to reviewers’ comments

Reviewer #1: The PK and tissue distribution of bromelain were conducted and compared in rats following oral and nasal administration, and the pharmacodynamic profile was determined in guinea pigs. The present study is within the scope of PLOS ONE. HOWEVER, the writing elements need further improvement to enhance the quality of the manuscript, and copy editing is beyond the scope of a reviewer’s charge. The authors should pay more attention to further polishing this manuscript. Some of my specific comments are listed below:

Comment 1: The animals used in this study (rats and guinea pigs) should be added to both the title and abstract. Otherwise, readers will be misled into thinking that this study was conducted on human patients.

Response: In order to prevent the wrong conception about the study by readers, correction in the title and abstract has been made as per reviewer’s suggestion. Title has been changed to “Implications of nasal delivery of bromelain on its pharmacokinetics, tissue distribution and pharmacodynamic profile- a preclinical study.” 

Similarly, changes has been made in abstract in revised manuscript i.e., “The present study investigated the preclinical prospects of nasal administration of bromelain on systemic bioavailability, tissue distribution and it’s in vivo anti-histaminic, bronchodilator and anti-asthmatic activity in animal models.” 

Comment 2: 2. 2.3. Pharmacokinetic studies: To this reviewer’s knowledge, six rats were not enough in a PK study. Additionally, where were the blood samples collected? How many milliliters of blood were collected each time? All these details should be given in the revision.

Response: As per reviewer’s suggestion, twelve Wistar rats per group were taken in PK study. Blood samples (300 µl) were collected from rat tail vein in heparinised coated tubes at pre-dose (0.0), 0.5, 1, 2, 4 and 6 h post dose of bromelain solution. 

These details have been in incorporated in revised manuscript as “Twenty four Wistar rats were randomly distributed into two groups with twelve animals in each group. Animals of group I received bromelain solution orally (400 µl, 40 mg/Kg body weight) by oral gavage while second group received 30 µl, bromelain solution (10 mg/ Kg body weight) in each nostril in 30 s with the help of micropipette with 0.1 mm internal diameter at delivery place. Blood samples (300 µl) were collected from rat tail vein in heparinised coated tubes at pre-dose (0.0), 0.5, 1, 2, 4 and 6 h post dose of bromelain solution. 

Comment 3: Three animals per group were sacrificed at 0.5, 1, 2, 4 and 6 h post dosing. Likewise, three rats per time point were not sufficient. These data may not be sufficient to describe inter-individual differences.

Response: As per reviewer’s suggestion, number of animals was increased to six at each time point i.e., six animals per group were sacrificed at 0.5, 1, 2, 4 and 6 h post dosing. The results of tissue distribution study at each time point is average of six readings obtained from six animals sacrificed at respective time. 

The corrections have been incorporated in revised manuscript under Tissue distribution study as “Sixty healthy wistar rats were randomized into two groups (n= 30). Rats of group I received bromelain solution orally (40 mg/kg body weight) while group II received bromelain solution nasally (10 mg/kg body weight) in each nostril with the help of micropipette with 0.1 mm internal diameter respectively. Six animals per group were sacrificed at 0.5, 1, 2, 4 and 6 h post dosing.

Comment 4: Animals were exposed to an aerosol of … The authors should pay more attention to English.

Response: As per reviewer’s suggestion, correction has been incorporated in revised manuscript i.e., “Animals were challenged with histamine dihydrochloride aerosol (0.5% w/v) using a nebulizer at a pressure of 300 mm Hg in an air tight Plexiglass chamber (24 × 14 × 24 cm) for 1 min. Consequently after histamine exposure, animals were closely observed to record the pre-convulsive dyspnea time and convulsion time for 9 minutes.”

Comment 5: Table 1: Which parameters were statistically significantly different (P<0.05)?

Response: All the pharmacokinetic parameters like Cmax, Tmax, ke, t1/2, MRT and AUC0-24h were statistically significantly different (p<0.05). The same has been highlighted in Table 1.

Comment 6: Figure 1B: This reviewer suggests the authors change the style of Figure 1B to Figure 1A.

Response: As per reviewer’s suggestion, style of Figure 1B has been changed to Figure 1A.

Reviewer #2: 

Comment 1: Abstract – The meaning of “unforeseeable prophylactic hypersensitivity due to its denaturation or agglomeration hinders its oral use” is unclear.

Response: In order to have more clarity changes has been made in abstract as per reviewer’s suggestion.

The changes made are presented in revised manuscript as “However, poor stability in gastric milieu, high dose and immunogenicity associated with prolonged use hinders its oral use.”

Comment 2: Section 2.4 – For the oral dose of bromelain solution (40 mg/kg), what is the volume of administration? Is the drug in a solution or in a suspension? What is the vehicle of the solution? Is there any pH adjustment?

The same questions can be applied to the nasal dose solution (10 mg/kg).

How is the nasal dose administered? Are the animals constrained in a chamber? Is the nasal dose administered over a duration?

Response: For the oral dose of bromelain solution (40 mg/kg), 400 µl of bromelain solution was administered by oral gavage. The vehicle of solution was distilled water. No pH adjustment of bromelain solution was done during study. 

30 µl of bromelain solution at a dose of 10 mg/Kg was administered in each nostril in 30 s with the help of micropipette with 0.1 mm internal diameter at delivery place. Nasal dose was administered using intransal grip, each rat was restrained twice and held with their neck parallel to the floor while administering nasal dose.

As per reviewer’s suggestion changes have been made in manuscript and represented as “Animals of group I received bromelain solution orally (400 µl, 40 mg/Kg body weight) by oral gavage while second group received nasally (30 µl, 10 mg/ Kg body weight) in each nostril in 30 s with the help of micropipette with 0.1 mm internal diameter at delivery place. Blood samples (300 µl) were collected from rat tail vein in heparinised coated tubes at pre-dose (0.0), 0.5, 1, 2, 4 and 6 h post dose of bromelain solution. Nasal dose was administered using intransal grip, each rat was restrained twice and held with their neck parallel to the floor while administering nasal dose. 

Comment 3: Table 1, the increased in AUC looks like to be caused by a decrease in elimination? It may not be due to increase in bioavailability alone. The improved in efficacy may be due to higher drug concentration in the lung at 0.5 hour.

Response: The decrease in elimination rate constant increased the t1/2 and mean residence time which contributed to increase in AUC i.e., systemic bioavailability. Nasal administration improved the drug concentration in lungs which might have improved therapeutic efficacy of bromelain. 

Comment 4: Figure 1 – it looks like the substantial difference in the lung concentration is only observed at 0.5 hour. In Figure 1, labelling with “A” and “B” and “a” and “b” is confusing.

Response: As per reviewer’s suggestion, labeling has been changed in Figure 1.

---

## [Decision Letter · Decision Letter 1]

4 Nov 2022

Implications of nasal delivery of bromelain on its pharmacokinetics, tissue distribution and pharmacodynamic profile- a preclinical study

PONE-D-22-16222R1

Dear Dr. Sharma,

We’re pleased to inform you that your manuscript has been judged scientifically suitable for publication and will be formally accepted for publication once it meets all outstanding technical requirements.

Kind regards,

Lisa Ann Tell

Academic Editor

PLOS ONE

Additional Editor Comments (optional):

Reviewers' comments:

Reviewer's Responses to Questions

**Comments to the Author**

1. If the authors have adequately addressed your comments raised in a previous round of review and you feel that this manuscript is now acceptable for publication, you may indicate that here to bypass the “Comments to the Author” section, enter your conflict of interest statement in the “Confidential to Editor” section, and submit your "Accept" recommendation.

Reviewer #1: All comments have been addressed

2. Is the manuscript technically sound, and do the data support the conclusions?

Reviewer #1: Yes

3. Has the statistical analysis been performed appropriately and rigorously? 

Reviewer #1: Yes

4. Have the authors made all data underlying the findings in their manuscript fully available?

Reviewer #1: Yes

5. Is the manuscript presented in an intelligible fashion and written in standard English?

Reviewer #1: Yes

6. Review Comments to the Author

Reviewer #1: The author has fully addressed my previous comments. I think this manuscript is acceptable in its current state.

7. PLOS authors have the option to publish the peer review history of their article (what does this mean?). If published, this will include your full peer review and any attached files.

Reviewer #1: **Yes: **Fan Yang

---

## [Editor Report · Acceptance letter]

10 Nov 2022

PONE-D-22-16222R1 

Implications of nasal delivery of bromelain on its pharmacokinetics, tissue distribution and pharmacodynamic profile- a preclinical study 

Dear Dr. Sharma:

I'm pleased to inform you that your manuscript has been deemed suitable for publication in PLOS ONE. Congratulations! Your manuscript is now with our production department. 

Kind regards, 

on behalf of

Dr. Lisa Ann Tell 

Academic Editor

PLOS ONE